# The Health-Promoting and Sensory Properties of Tropical Fruit Sorbets with Inulin

**DOI:** 10.3390/molecules27134239

**Published:** 2022-06-30

**Authors:** Agnieszka Palka, Magdalena Skotnicka

**Affiliations:** 1Department of Quality Management, Faculty of Management and Quality Science, Gdynia Maritime University, 81-225 Gdynia, Poland; 2Department of Commodity Science, Faculty of Health Science, Medical University of Gdansk, 80-210 Gdansk, Poland; skotnicka@gumed.edu.pl

**Keywords:** tropical fruit, sorbet, antioxidative activity, polyphenols, inulin

## Abstract

Inulin is a popular prebiotic that is often used in the production of ice cream, mainly to improve its consistency. It also reduces the hardness of ice cream, as well as improving the ice cream’s organoleptic characteristics. Inulin can also improve the texture of sorbets, which are gaining popularity as an alternative to milk-based ice cream. Sorbets can be an excellent source of natural vitamins and antioxidants. The aim of this study was to evaluate the effect of the addition of inulin on the sensory characteristics and health-promoting value of avocado, kiwi, honey melon, yellow melon and mango sorbets. Three types of sorbets were made—two with inulin (2% and 5% wt.) and the other without—using fresh fruit with the addition of water, sucrose and lemon juice. Both the type of fruit and the addition of inulin influenced the sorbet mixture viscosity, the content of polyphenols, vitamin C, acidity, ability to scavenge free radicals using DPPH reagent, melting resistance, overrun and sensory evaluation of the tested sorbets (all *p* < 0.05). The addition of inulin had no impact on the color of the tested sorbets, only the type of fruit influenced this feature. In the sensory evaluation, the mango sorbets were rated the best and the avocado sorbets were rated the worst. Sorbets can be a good source of antioxidant compounds. The tested fruits sorbets had different levels of polyphenol content and the ability to scavenge free radicals. Kiwi sorbet had the highest antioxidant potential among the tested fruits. The obtained ability to catch free radicals and the content of polyphenols proved the beneficial effect of sorbets, particularly as a valuable source of antioxidants. The addition of inulin improved the meltability, which may indicate the effect of inulin on the consistency. Further research should focus on making sorbets only from natural ingredients and comparing their health-promoting quality with the ready-made sorbets that are available on the market, which are made from ready-made ice cream mixes.

## 1. Introduction

Sorbets are low in calories (60–120 kcal/100 g of the product), mainly due to the lack of added fat, milk, cream and animal protein [1]. Due to the high water content, they are a very light and refreshing dessert, eagerly eaten, especially in the summer months when the air temperature is high. Moreover, they are a suitable type of ice cream for people suffering from allergies or intolerance to the ingredients of milk-based ice cream. Sorbets are made of fruits and/or vegetables, juices and water, with the addition of sweeteners and stabilizers; they should have an attractive taste, low energy value and be easily digestible.

Inulin is a natural prebiotic. Prebiotic was described as “a non-digestible food ingredient that beneficially affects the host by selectively stimulating the growth and/or activity of one or a limited number of bacteria in the colon, and thus improves host health” [2]. This definition was almost unchanged for more than 15 years. According to this definition, inulin can be classified as a prebiotic due to its selective promotion of the activity and growth of particular native bacteria of the digestive tract [2,3,4]. Its addition to ice cream desserts increases their health value and gives the products the features of functional food. As a fat substitute, inulin lowers the energy value of the product [5]. It can be a texturing agent, improving the consistency, stabilizing and thickening, and it can also affect the taste and smell of products. Sorbets can be a valuable carrier of prebiotics in the diet [6,7,8]. Health-promoting additives that are introduced into sorbets should increase the sensory quality of the product or not significantly affect it. Inulin that occurs in different parts of plants is resistant to digestion in the human digestive tract and is fermented in the colon [3]. Food industries are commercializing the use of inulin as a food ingredient due to its functional and physicochemical properties, i.e., melting point, gelation property, glass transition temperature and gel integrity [3]. The viscosity of inulin-containing solutions is directly linked with its concentration. A little increase in viscosity is observed when the concentration of inulin is increased from 1 to 10%, followed by a continuous increase in thickness with the increase in concentration up to 30%, but with no obvious signs of gel formation. Inulin can alter the composition of intestinal microflora and stimulate the growth of beneficial bacteria, as well as inhibit the growth of potentially pathogenic bacteria [3,9].

The speed at which the ice cream mix freezes depends on the water, sugar, fat and protein content of the ice cream. In the case of sorbets, it is mainly the ratio of water to carbohydrates. The sugar content of the fruit is therefore important, as is the amount of sugar that is added to the ice cream mixture. The density and viscosity of the mixture are significant parameters [1,10,11,12].

Blanching is commonly used as a pretreatment to freezing, although its application is limited in fruits when compared with vegetables [13]. Pasteurization and other thermal treatments reduce the content of vitamins and polyphenols [14,15]; therefore, heating fruit sorbet mix before freezing it should be avoided. Heat processing changed the intensity of the aromatic sensorial profiles of mango purees from three Thai varieties in Wongkaew et al.’s [16] research.

Freezing allows seasonal fruits and their products to be available to consumers throughout the year. Freezing temperatures crystallize water, which accounts for approximately 85–90% of the fruit [13]. Frozen fruits are preserved by the implementation of negative temperatures; however, slow freezing, an effect of the large ice crystals that are present in the interstices, destroys the cellular integrity of fruits, because the initial rigidity of the outer cellulosic wall depends on the state of the pectic cement, which is linked to the maturity of the fruits [15]. Therefore, fast freezing should be applied in ice cream and sorbet production. Scraped surface heat exchangers (SSHEs) are commonly used in ice cream production, where the purpose of the SSHE is to chill and freeze a formulated mix in order to crystallize the water that is present on the walls of the inner pipe [1]. The blades also serve in this case as a dispersion system for the ice that is formed and a certain amount of air that could be introduced into the product. Air bubbles and ice crystals sizes are major quality parameters in sorbets, and their occurrence is closely related to the heat transfer mechanism and the flow characteristics in the SSHE [17].

The term “superfood” refers to foods that are beneficial to human health due to their high levels of nutrients and bio-active phytochemicals, such as antioxidants [18]. Fruits are widely recognized as an excellent source of bioactive phenolic compounds [19,20,21]. Polyphenols play important roles in the body as antioxidants and/or cellular messengers. Avocado fruit (*Persea americana*) is a high fat fruit, contains rare sugars of high carbon number and is relatively rich in certain vitamins, dietary fiber, minerals and nitrogenous substances. It is a rich source of potassium, sodium, magnesium, vitamin A, vitamin C, vitamin B6, niacin, pantothenic acid, riboflavin, choline, lutein/zeaxanthin, cryptoxanthin, phytosterols and high-monounsaturated fatty acids (MUFA), and it contains lesser amounts of biotin, folic acid, thiamin, riboflavin, vitamin D, vitamin E and vitamin K [22,23,24]. Cantaloupe (*Cucumis melo*) is one of the most consumed fruits worldwide because of its flavor and health benefits. Cantaloupe is an excellent source of vitamins A and C, potassium and magnesium, and has useful medicinal properties, e.g., antioxidant, anticancer, antimicrobial, antidiabetic and hepatoprotective effects, as well as activity against hypothyroidism and immune-modulator action [25,26,27,28,29]. Kiwifruit (*Actinidia*) is a nutrient-dense fruit that is exceptionally high in vitamin C and contains nutritionally relevant levels of dietary fiber, potassium, vitamin E and folate, as well as various bioactive components, including a wide range of antioxidants, phytonutrients and enzymes that provide functional and metabolic benefits. The kiwifruit (*A. deliciosa* and *A. chinensis*—“green” and “gold” cultivars, respectively) of commercial cultivation are large-fruited selections of predominantly green kiwifruit and an increasing range of gold varieties [30]. The flesh of the green Hayward cultivar is described as tangy, sweet and sour, a unique flavor combination, whereas the gold cultivar is described as having a sweet and tropical taste [30,31,32]. The mango (*Mangifera indica* L.) is a tropical fruit, originally from the South of Asia, and it is available worldwide today. Mango is rich in vitamin A and contains reasonable amounts of vitamins B and C, as well as minerals, mainly calcium, phosphorus and iron. Mango fruits provide energy, dietary fiber, carbohydrates, proteins and fats. Mango is also a particularly rich source of polyphenols [33,34,35]. Mangoes have a short shelf-life and are often processed to facilitate exportation and to preserve the fruit past its season. The main and directly consumable part of the mango is the pulp, which accounts for 50 to 60% of the total weight of the fruit and is used to prepare various products such as juice, jam, puree and nectar [16,33,34,36,37]. The aim of the work was to produce tropical fruit sorbets with and without the addition of inulin, and evaluate the impact of the fruit and inulin on selected health-promoting and sensory properties of these sorbets.

## 2. Materials and Methods

### 2.1. Raw Materials for Sorbets

The basic ingredients of the sorbets were fresh fruits which were obtained at edible ripeness from a local grocery store. The choice of these fruits was guided by popularity and their worldwide availability, as well as by their health value, due to being excellent sources of vitamins and microelements, and having medicinal properties, such as analgesic, anti-inflammatory, antioxidant, antiulcer, anticancer, antimicrobial, diuretic, and antidiabetic properties [29,38,39]. The research material was sorbets that were made of selected tropical fruits: avocado, kiwifruit, 2 varieties of melon (cantaloupe and honeydew) and mango with the addition of water, sucrose and lime juice. Three types of products were produced: with the addition (fruit replacement) of inulin (2 and 5% wt.) and without inulin. Chicory inulin was purchased in a local health food store. According to the information given on the package, the product was produced in Belgium and contained 89 g of fiber per 100 g of product.

### 2.2. Preparation of the Sorbets

A total of 15 sorbets were made due to the addition of different fruits and inulin. The sorbets with inulin were prepared by replacing 2% or 5% fruits with inulin. Fruits (without peels and seeds) with the addition of water, sugar, lemon juice and inulin (Table 1) were mixed in a Thermomix TM5. The sorbets mixes were prepared as given in Table 1. Each sorbet mixture was mixed separately, stored at 4 °C for 24 h and then frozen in Ice Cream Machine Unold 48840. The sorbets were stored at −25 °C in the laboratory of the Gdynia Maritime University and evaluated after two days of storage.

### 2.3. Physico-Chemical Parameters

The viscosity of the ice cream mixture was determined by a Fungilab Viscolead Pro Viscometer VL321003.

The melting resistance was measured by determining the number of melted sorbets from a given sample volume. The method consisted in determining the volume as a result of sorbet leachate after 60 min at room temperature. This is an indicator of the resistance of ice cream to melting. Procedure of the determination: cylinder-shaped samples were taken from the frozen ice cream using a metal cylinder-shaped mold with a capacity of 24.73 cm^3^. The entire mass of the mold was transferred to a sieve that was mounted on a funnel. The funnel was placed in a measuring cylinder. After 60 min, the volume of leakage was read from the cylinder. The test was carried out at a temperature of 20 °C.

The calculations were obtained according to the formula:(1)$$V=\frac{V1}{V2}\times 100\%$$
where:*V*—melting resistance [%]*V*1—volume of melted sorbet [cm^3^]*V*2—volume of cylinder (24.73) [cm^3^]

The overrun was measured by determining the amount of air in a given volume of the sample. A defined volume of the sample (cut from the product with a cylinder) was transferred to a volumetric flask and made up to the mark with distilled water. Based on the known volume of the volumetric flask, the volume of the cylinder, and the amount of added water, the overrun of the sorbets was calculated.

The total phenolic content (TPC) was determined using Folin–Ciocalteu reagent and a standard curve of Gallic acid (AR), and the results were expressed as mg Gallic acid equivalent (GAE)/100 g sample [40].

The antioxidant activity was estimated as free radical scavenging activity using 1,1-diphenyl-2-picryl-hydrazil (DPPH) [41].

The vitamin C content was determined by Tillmans’ method [42].

Titratable acidity was determined according to the AOAC method [43].

All analyses were conducted three times.

### 2.4. Sensory Evaluation

The sensory assessments were performed by 10 semi-trained judges (7 female, 3 male, age 30–49, employees of the Department of Quality Management, Faculty of Management and Quality Science, Gdynia Maritime University, Gdynia, Poland). The coded samples were served to panelists under normal daylight. Each panelist received samples at approximately −12 °C for evaluation (this temperature is recommended while serving ice cream). A 5-point hedonic scale was employed for the evaluation of color, odor, taste, consistency and overall preference of sorbets (Table 2). After receiving the evaluation results of all products, the average values of the individual evaluations were calculated.

### 2.5. Statistical Analysis

A two-way analysis of variance (ANOVA) was used for the statistical analysis. The type of fruit and the effect of the added inulin was considered. Calculations were made using Microsoft Excel MSO 2016 (ver. 2205, Poland) and Statistica 13 software (ver. 13.1.336.0 StatSoft, Poland). The post hoc Tukey’s procedure was used to find patterns and relationships between the subgroups. Differences among the groups were determined as statistically significant at a level of *p* ≤ 0.05.

## 3. Results

The first assessed feature is the appearance and color of the product, then the smell is assessed, followed by the taste and consistency. Additionally, the panelists were asked to give an overall rating (the “overall preference”). The odor of the avocado sorbets was not significantly different to the cantaloupe sorbets. The taste of the avocado with inulin sorbet was not significantly different to the cantaloupe sorbets, and in addition was not significantly different to the sorbets of kiwi and mango. Despite the high health value, these avocado features in terms of sorbet production were not high. The best rated was the mango sorbet with the addition of inulin, which could have contributed to the improvement of the consistency of the product (Table 3). The consistency of this sorbet was rated the highest. The smell of the melon sorbets was rated at 3.5–4.0 points, while their consistency was rated the worst. This could be because the melons contain a lot of water and all the mixtures in the experiment were prepared in the same way. The amount of water that was added into the sorbet mixtures influenced the sensory evaluation, as well as the melting resistance of the sorbets. The color of the mango sorbets was not significantly different to the kiwi with inulin sorbet, and these products were rated the best. Inulin addition did not influence the color estimation. In overall preference, the mango and kiwifruit sorbets were evaluated as the best products. Regarding the overall preference, the sorbets of mango, kiwi, honeydew melon, and cantaloupe melon were statistically equal, although the mango and kiwifruit sorbets were evaluated with the highest points estimation. The addition of inulin generally improved all the features that were estimated in the organoleptic assessment. A slight odor deterioration of the cantaloupe melon sorbets with inulin was found; nevertheless, the difference of odor estimation was statistically not significant. The addition of inulin (both 2 and 5% wt.) did not statistically affect the organoleptic features of the sorbets from the given fruit.

The melting resistance of the tested sorbets ranged from 11.46 to 38.41%, and the addition of inulin influenced this parameter, depending on the type of fruit and the amount of the additive (Table 4). The addition of 5% of inulin influenced the meltability of all the sorbets, while the addition of 2% of inulin affected the melting resistance of the kiwifruit sorbet. The highest melting resistance values were obtained by the kiwifruit sorbets, as well as the melon sorbets, which could have been caused by the high water content in these fruits. Inulin addition decreased the melting resistance of the sorbets. Ice cream overrun is influenced by the protein and fat content. The overrun of sorbets is therefore lower than the overrun of milk-based ice cream and is usually at the level of 10–40%. This parameter is influenced by the composition of the mixture. In the tested sorbets, the overrun ranged from 12.33 to 19.67%, which was influenced by both the type of fruit and the inulin addition. The addition of inulin influenced the overrun of cantaloupe melon and mango sorbets, but the amount of added inulin did not significantly influence that parameter. The amount of added inulin significantly influenced the overrun of the kiwifruit sorbets, whereas only 5% of added inulin influenced the overrun of the yellow honeydew melon sorbet. This parameter was not influenced in the avocado sorbets. The total polyphenols in the sorbets ranged from 4.83 to 10.97 (mg GAE/g product), depending on the type of sorbet. The highest number of total polyphenols was estimated in the avocado sorbets, and the lowest in the honeydew melon sorbets. The kiwifruit sorbets were also high in total polyphenols. The addition of inulin influenced this parameter in the avocado and mango sorbets, but this was probably due to the smaller amount of fruit that was used in the production of the product. The DPPH assay showed the highest antioxidative activity in the kiwifruit and mango sorbets, while the lowest antioxidative activity was assessed in the sorbet with the lowest content of total polyphenols (honeydew melon). The largest amount of vitamin C was determined in the kiwifruit and mango sorbets, and the lowest in the melon sorbets. The inulin sorbets contained lower amounts of vitamin C, total polyphenols and oxidative activity. The determination of titratable acidity showed that the acidity of all the sorbets was in the range of 2.2–4.5°SH; the kiwifruit sorbet showed the highest acidity, and the addition of inulin increased or decreased this parameter, depending on the fruit.

The ANOVA analysis of variance showed an influence or no influence of the type of fruit and the addition of inulin on the examined features. The type of fruit had a significant impact on all the examined discriminants (*p* = 0.0000). The addition of inulin had no significant effect only on the acidity of the tested sorbets (*p* = 0.0949), while fruit plus inulin had a significant effect on the content of polyphenols (*p* = 0.0031), vit. C (*p* = 0.0000) and the acidity of sorbets (*p* = 0.0001). In the case of the sensory evaluation, the statistical analysis showed that the type of fruit influenced all the assessed characteristics (*p* = 0.0000), while the addition of inulin influenced the taste of the sorbets (*p* = 0.0187), their consistency (*p* = 0.0242) and overall preference (*p* = 0.0149). The consistency of the product affects the mouthfeel and overall preference, and in sorbets, it is highly influenced by the time of freezing and the viscosity of the ice cream mixture. The addition of inulin, both 2 and 5%, statistically influenced the viscosity of the prepared sorbet mixtures. The addition of inulin increased the viscosity, and the greater the addition, the more the viscosity increased.

The addition of inulin (2% wt.) to the avocado sorbet (replacing 2% wt. of fruit with inulin) only influenced the viscosity of the prepared mix, while 5% of inulin in the mixture had a statistically significant effect on the improvement of the melting resistance; however, the total polyphenols content and antioxidative properties of this sorbet significantly decreased. The addition of inulin (both 2 and 5% wt.) increased the viscosity of all the sorbets. Inulin in the cantaloupe melon sorbets improved the overrun and melting resistance but decreased the acidity. Inulin addition improved the melting resistance of the kiwifruit, mango and yellow honeydew melon sorbets. The mango sorbets with inulin had lower total polyphenols content, probably due to the lower fruits content, and in these sorbets the difference was significant.

## 4. Discussion

The quality of fruits and their products are influenced by four characteristics: color/appearance, flavor, texture, and nutritional value. Consumers first evaluate the color and appearance of the fruit before deciding whether to eat it, and then its flavor helps to determine if they will consume it again [44]. The same principle applies to many products, especially fruit and vegetable preserves. Products such as sorbets must have an acceptable, attractive color. Sorbets should be made of fresh and good quality fruits or vegetables. Consumers are willing to pay more for high-quality products, comparable to the fresh and minimally processed items [13]. The level of fruit ripeness is an important key to ensuring an appropriate level of quality, both sensory and pro-health. Harvesting and storage conditions are crucial determinants of the quality of the fruit food materials that are used fresh and in various functional-type food products and supplements [45]. Despite those factors preceding sorbet preparation, freezing process conditions can affect the flavor, chemical composition, and the color of fruit products. Nevertheless, freezing is recognized as the postharvest technique that best preserves fruit flavor.

Consumption of tropical fruits, such as melons, kiwifruit, avocado or mango, can help fight several diseases. Phytochemicals can fight human diseases and help in preventing cancer, fighting depression, preventing ulcers and removing dandruff, as well as stimulating the immune system [46,47]. However, the vitamins and polyphenol concentration vary not only in different plant species, but also between cultivars, and may be affected by harvesting and storage conditions [48].

Freezing is regarded as a technique that has little damage effect on the phenolic content of fruits. Some authors reported an increase in the concentration of phenolic compounds after freezing [49], while other studies have shown a significant reduction. Freezing can reduce the concentration of phenolics with no effect on the total antioxidant capacity of the juice, due to the high stability of L-ascorbic acid. This difference could be a consequence of the thawing process [50,51]. Frozen storage for one year did not cause significant losses in the content of polyphenolic antioxidants in most of the analyzed fruits (strawberries, raspberries, red currants, cherries, sour cherries, hawthorn, cornelian cherries, black-berries, white and red grapes) [52]. The investigated sorbets had high antioxidant activity, although the amount of total polyphenols was at a low level (5–11 mg GAE/g product). The avocado and mango sorbets that were studied by Palka [53] contained much higher total phenols (over 260 mg GAE/g product). Such a difference could have been due to the different quality of the fruit that was purchased for research. This could be further evidence that the initial quality of fruits is crucial in products with health-promoting features of functional food.

Freezing usually leads to losses in vitamin C content. These losses depend both on the method of freezing (time and temperature of the process) and on the storage conditions (temperature and its fluctuation) A reduction in vitamin C content is linked to drip losses after freezing, and not to a true degradation process [54].

The total vitamin C content is an attribute of kiwifruit (80 and 120 mg/100 g fresh weight Hayward green cultivar, while 18 mg/100 g is given for melon and 27.7 mg/100 g for fresh mango). This natural variation in the amount of vitamin C in fruit, including kiwifruit, is due to numerous factors, including growing region and conditions, time and maturity at harvest, and storage conditions [32,55]. The vitamin C content in the sorbets was significantly lower (<0.5 mg for melon sorbets, <9.5 mg for mango sorbets and <15 mg for kiwifruit sorbets) due to a lower content of the fruits in the products, but also due to the low initial content of vitamin C in the fruits that were used in the sorbets’ production. Therefore, it is important to use good quality fruits in the production of fruit preserves.

The lower content of vitamin C and polyphenols in the tested sorbets with the addition of inulin could be the result of the lower content of fruit in the mixture. The addition of lemon juice to all the sorbets could also have affected the results that were obtained.

The flavor of mango cultivars changes during the production of sorbet; therefore, sorbet manufacturers should select cultivars based on their properties in sorbet. To assist in selecting cultivars for mango puree and sorbet production, consumer studies should be conducted to determine which attributes influence preference [36]. The flavor of sorbets is influenced not only by selected fruits, but also by the addition of sugar, lemon, orange or lime juice. The flavor can also be influenced by the addition of inulin.

Pintor et al. [56] investigated the effect of inulin on the melting resistance and textural properties of low-fat and sugar-reduced ice cream. In their research, higher apparent viscosity resulted in a more stable system with a higher overrun, where inulin controlled the available water. The improvement in melting properties reflected the stable state of the air bubbles-emulsified fat-ice crystals matrix, where the putative effect of inulin to retain water compensating solids and fat reduction retarded the melting of the ice crystals. Inulin retained free water when butyric fat and sugar were reduced, resulting in smaller ice crystals, reflecting a softer texture. Inulin is a functional ingredient (soluble fiber and prebiotic) and can be employed to reduce 30% of the butyric fat content and 12% of the sugar content in the formulation of low-fat reduced-sugar ice cream [56]. The addition of inulin significantly influenced the acidity and color parameters of sorbets in Przybylski et al.’s studies [9]. The content of inulin in frozen ice cream desserts should be determined individually, depending on the type of ice sorbet. The results of the research show that the addition of inulin to fruit and vegetable ice sorbets allows for obtaining products of a satisfactory sensory quality; good physicochemical properties impacted most of the studied discriminants.

Due to many studies proving the high pro-health value of fruit peels, it is also advisable that, in order to reduce waste, fruit with peel should be used for the production of sorbets. Several studies have proven the presence of a wide range of bioactive compounds in various fruit industrial by-products, which are essentially pomace, peels and seed fractions. These compounds consist mainly of carbohydrates, secondary metabolites, lipids and proteins. Generally, seeds are rich in polyphenols and bioactive lipids, whereas peels are considered as a rich source of dietary fibers. Bioactive compounds are present in fruit by-products with various concentrations and combinations [2,39,57,58]. Avocado peel has antioxidant activity and the potential to develop as a functional food [59]. In this study, all the fruit peels were removed; therefore, where possible, the removal of fruit peel should be considered when manufacturing sorbets with peels. Some peels can be difficult to crush into small enough pieces that are not detectable in an organoleptic evaluation; therefore, this method of preparing the mixture is crucial.

Palka [60] verified consumers’ opinion of sorbets that were produced from three old apple varieties; all the apples were processed into sorbets, including the peel. The sensory characteristics of traditional varieties were well accepted by consumers [60]. The same author [53] studied the best proportions of ingredients to produce sorbets based on avocado, with a mango flavoring and color enhancer. The organoleptic assessment and physicochemical characteristics of the sorbets confirmed that it was possible to create sensory-attractive sorbets with pro-health properties [36]. Avocado improves the bioavailability of nutrients from other plant-based foods. Therefore, consuming avocados with other fruit and vegetables can be beneficial to human health [22].

The amount and type of sweeteners that are added have a large influence on the consistency and hardness of ice cream and sorbets. The addition of sucrose causes a harder and more brittle consistency; therefore, in addition to sucrose, glucose and maltodextrin are most often added to ice cream, which improves its consistency. The improvement in consistency by adding other sucrose substitutes can also be achieved by using xylitol. Naknaen and Itthisoponkul studied the influence of xylitol on the texture of cantaloupe jam. The increase in the amount of xylitol caused a decrease in the hardness of the jam. The increase in xylitol content causes an apparent decrease in hardness, meaning xylitol could be used as a substitute for sucrose as a low-calorie sweetener [27]. The use of various fruits, as well as various sweeteners, can improve the health-promoting properties of sorbets. These properties are also influenced by the times and techniques of freezing. Pavlyuk et al. [61] scientifically substantiated and experimentally proved the possibility of using the cryogenic “shock” freezing and cryomechanolysis methods for the finely dispersed shredding of fruits and vegetables as the innovative method for structure formation, and for obtaining fruit and vegetable ice cream-sorbets with a record content of BAS [61].

Considering the different starting compositions of the fruits that were used in the experiment, it is not surprising that the sorbets’ consistencies were different, and thus their sensory evaluation. In this study, it was proved that for each type of fruit, the ice cream mixture should be composed in such a way as to ensure the appropriate sensory experience, without a significant loss of the pro-health values.

## 5. Conclusions

The addition of inulin improved the consistency and melting resistance of the ice cream and influenced the physicochemical properties. The addition of 5% of inulin was better than a 2% addition. The overall preference of the assessed sorbets was rated as good (4.1) to almost very good (4.8) in the sensory evaluation, with the exception of the avocado sorbet. Many scientific reports claim that sorbets can be a valuable source of vitamins and polyphenols. The conducted research suggests that in the case of the tested fruits, a combination of different fruits should be prepared in order to obtain the excellent quality of the sorbet. Due to their high pro-health and technological value, avocados could be added to ice cream and sorbets that are made of other fruits, but they should not be used as a base for sorbets due to their sensory qualities. The addition of inulin (both 2 and 5% wt.) did not statistically affect the organoleptic features of the sorbets, but improved the physical features (mixture viscosity, overrun of finished products and melting resistance). Therefore, the addition of inulin to fruit sorbets could be replaced by other ingredients that could improve both the organoleptic and pro-health features of the sorbets, or the addition of inulin should replace the water in sorbets, instead of fruits.

## Figures and Tables

**Table 1 molecules-27-04239-t001:** The composition of the sorbets [%wt].

Type of Fruit/Name of the Product	Fruits Amount [%]	Water [%]	Sucrose [%]	Lemon Juice [%]	Inulin [%]
Avocado	48	40	10	2	0
Avocado with 2% of inulin	46	40	10	2	2
Avocado with 5% of inulin	43	40	10	2	5
Cantaloupe melon	48	40	10	2	0
Cantaloupe melon with 2% of inulin	46	40	10	2	2
Cantaloupe melon with 5% of inulin	43	40	10	2	5
Kiwifruit	48	40	10	2	0
Kiwifruit with 2% of inulin	46	40	10	2	2
Kiwifruit with 5% of inulin	43	40	10	2	5
Mango	48	40	10	2	0
Mango with 2% of inulin	46	40	10	2	2
Mango with 5% of inulin	43	40	10	2	5
Honeydew melon (yellow)	48	40	10	2	0
Honeydew melon (yellow) with 2% of inulin	46	40	10	2	2
Honeydew melon (yellow) with 5% of inulin	43	40	10	2	5

**Table 2 molecules-27-04239-t002:** 5-point hedonic scale for sensory evaluation.

Score	Organoleptic quality				
	Overall preference	Color	Odor	Taste	Consistency
1	Dislike	Dark	Dislike	Very poor	Very poor
2	Neither like	Slightly dark	Neither like	Poor	Poor
3	Like slightly	Moderate	Like slightly	Fair	Fair
4	Like moderately	Pale	Like moderately	Good	Good
5	Like very much	Very pale	Like very much	Very good	Very good

Additional information: (1) untypical for used fruits; (2) untypical for used fruits; (3) slightly typical; (4) typical for used fruits; (5) very typical for used fruits.

**Table 3 molecules-27-04239-t003:** Sensory evaluation of the sorbets.

Type of Sorbet	Addition of Inulin [%]	Overall Preference	Color	Odor	Taste	Consistency
		$\bar{X}$ ± SD	$\bar{X}$ ± SD	$\bar{X}$ ± SD	$\bar{X}$ ± SD	$\bar{X}$ ± SD
Avocado	0	3.0 ± 0.471 ^a^	4.0 ± 0.471 ^a^	2.7 ± 1.160 ^a^	2.8 ± 1.033 ^a^	4.4 ± 0.516 ^abcd^
Avocado	2	3.5 ± 0.527 ^ab^	4.1 ± 0.738 ^ab^	3.4 ± 0.843 ^ab^	3.3 ± 0.823 ^ab^	4.4 ± 0.516 ^abcd^
Avocado	5	3.7 ± 0.483 ^abc^	4.2 ± 0.632 ^abc^	3.5 ± 0.707 ^ab^	3.4 ± 0.699 ^abc^	4.2 ± 0.422 ^abc^
Cantaloupe melon	0	4.2 ± 0.632 ^bcd^	3.8 ± 0.422 ^a^	4.0 ± 0.471 ^b^	4.1 ± 0.876 ^bcde^	3.7 ± 0.483 ^a^
Cantaloupe melon	2	4.3 ± 0.483 ^bcd^	4.1 ± 0.316 ^ab^	3.8 ± 0.422 ^b^	4.3 ± 0.483 ^cde^	3.9 ± 0.568 ^ab^
Cantaloupe melon	5	4.4 ± 0.516 ^cd^	4.1 ± 0.316 ^ab^	3.9 ± 0.316 ^b^	4.3 ± 0.483 ^cde^	4.2 ± 0.422 ^abc^
Kiwifruit	0	4.2 ± 0.632 ^bcd^	4.1 ± 0.568 ^ab^	3.9 ± 0.876 ^b^	3.9 ± 1.101 ^bcd^	4.1 ± 0.738 ^abc^
Kiwifruit	2	4.4 ± 0.699 ^cd^	4.3 ± 0.483 ^abc^	4.1 ± 0.994 ^b^	4.1 ± 0.568 ^bcde^	4.4 ± 0.699 ^abcd^
Kiwifruit	5	4.6 ± 0.516 ^d^	4.3 ± 0.483 ^abc^	4.2 ± 0.919 ^b^	4.2 ± 0.422 ^bcde^	4.6 ± 0.516 ^bcd^
Mango	0	4.6 ± 0.516 ^d^	4.8 ± 0.422 ^bc^	4.2 ± 0.422 ^b^	4.6 ± 0.516 ^de^	4.4 ± 0.516 ^abcd^
Mango	2	4.7 ± 0.483 ^d^	4.9 ± 0.316 ^c^	4.3 ± 0.483 ^b^	4.9 ± 0.316 ^e^	4.8 ± 0.422 ^cd^
Mango	5	4.8 ± 0.422 ^d^	4.9 ± 0.316 ^c^	4.4 ± 0.516 ^b^	4.9 ± 0.316 ^e^	5.0 ± 0.000 ^d^
Yellow honeydew melon	0	4.2 ± 0.632 ^bcd^	3.9 ± 0.568 ^a^	3.5 ± 0.527 ^ab^	3.9 ± 0.568 ^bcd^	3.9 ± 0.568 ^ab^
Yellow honeydew melon	2	4.1 ± 0.568 ^bcd^	3.9 ± 0.316 ^a^	3.5 ± 0.527 ^ab^	4.2 ± 0.422 ^bcde^	3.7 ± 0.483 ^a^
Yellow honeydew melon	5	4.3 ± 0.483 ^bcd^	3.9 ± 0.316 ^a^	3.6 ± 0.516 ^ab^	4.2 ± 0.422 ^bcde^	3.9 ± 0.316 ^ab^

Mean values ± standard deviations; n = 10; ^a,b,c (…)^—mean values followed by the different letters in the same line differ significantly at the significance level *p* ≤ 0.05.

**Table 4 molecules-27-04239-t004:** Physicochemical analysis of the sorbets.

Type of Sorbet	Addition of Inulin [%]	Acidity [°SH]	Vit. C [mg/100 g]	Total Polyphenols [mg GAE/g Product]	DPPH [%]	Overrun [%]	Melting Resistance [%]	Viscosity [cSt]
		$\bar{X}$ ± SD	$\bar{X}$ ± SD	$\bar{X}$ ± SD	$\bar{X}$ ± SD	$\bar{X}$ ± SD	$\bar{X}$ ± SD	$\bar{X}$ ± SD
Avocado	0	3.00 ± 0.000 ^ab^	1.80 ± 0.181 ^b^	10.97 ± 0.306 ^f^	49.87 ± 2.098 ^d^	17.33 ± 0.577 ^def^	20.89 ± 1.167 ^de^	15.04 ± 0.070 ^f^
Avocado	2	3.50 ± 0.500 ^bcd^	1.71 ± 0.163 ^b^	10.30 ± 0.346 ^ef^	47.63 ± 1.115 ^cd^	18.33 ± 0.577 ^fg^	18.70 ± 0.693 ^cd^	16.23 ± 0.199 ^g^
Avocado	5	3.50 ± 0.500 ^bcd^	1.73 ± 0.148 ^b^	9.90 ± 0.346 ^e^	47.17 ± 0.351 ^c^	18.67 ± 0.577 ^fg^	11.46 ± 1.167 ^a^	18.15 ± 0.242 ^h^
Cantaloupe melon	0	3.50 ± 0.500 ^bcd^	0.47 ± 0.015 ^a^	6.70 ± 0.346 ^b^	46.53 ± 0.306 ^c^	12.33 ± 0.577 ^a^	33.02 ± 1.167 ^i^	10.09 ± 0.043 ^a^
Cantaloupe melon	2	2.17 ± 0.289 ^a^	0.46 ± 0.040 ^a^	6.07 ± 0.321 ^b^	46.07 ± 0.231 ^c^	13.33 ± 0.577 ^ab^	26.96 ± 1.167 ^fg^	12.17 ± 0.062 ^c^
Cantaloupe melon	5	2.17 ± 0.289 ^a^	0.46 ± 0.031 ^a^	6.27 ± 0.058 ^b^	45.07 ± 1.026 ^c^	13.33 ± 0.577 ^ab^	23.30 ± 0.872 ^ef^	14.15 ± 0.145 ^e^
Kiwifruit	0	4.17 ± 0.306 ^de^	15.85 ± 0.493 ^e^	9.57 ± 0.115 ^d^	64.70 ± 1.200 ^e^	15.00 ± 1.000 ^bc^	38.41 ± 2.022 ^j^	11.14 ± 0.064 ^b^
Kiwifruit	2	4.27 ± 0.058 ^de^	14.32 ± 0.284 ^d^	9.57 ± 0.115 ^d^	63.97 ± 0.945 ^e^	15.67 ± 0.577 ^cd^	31.35 ± 1.083 ^hi^	12.13 ± 0.056 ^c^
Kiwifruit	5	4.27 ± 0.058 ^d^	14.68 ± 0.602 ^d^	9.63 ± 0.115 ^d^	63.70 ± 0.917 ^e^	16.00 ± 0.577 ^cde^	20.89 ± 1.173 ^de^	13.09 ± 0.194 ^d^
Mango	0	2.83 ± 0.289 ^ab^	9.21 ± 0.170 ^c^	7.57 ± 0.231 ^c^	64.97 ± 0.503 ^e^	17.67 ± 0.577 ^ef^	23.03 ± 0.929 ^e^	13.17 ± 0.053 ^d^
Mango	2	3.17 ± 0.289 ^bc^	8.76 ± 0.137 ^c^	6.10 ± 0.346 ^b^	63.70 ± 0.200 ^e^	18.67 ± 0.577 ^fg^	16.85 ± 1.167 ^bc^	15.26 ± 0.073 ^f^
Mango	5	3.17 ± 0.289 ^bc^	8.94 ± 0.211 ^c^	6.27 ± 0.058 ^b^	62.90 ± 1.039 ^e^	19.67 ± 0.577 ^g^	13.48 ± 1.167 ^ab^	18.10 ± 0.096 ^h^
Yellow honeydew melon	0	4.50 ± 0.000 ^e^	0.43 ± 0.018 ^a^	5.07 ± 0.252 ^a^	32.37 ± 0.252 ^b^	13.33 ± 0.577 ^ab^	34.37 ± 2.022 ^i^	10.20 ± 0.086 ^a^
Yellow honeydew melon	2	3.97 ± 0.058 ^cde^	0.39 ± 0.015 ^a^	4.83 ± 0.351 ^a^	29.47 ± 0.404 ^a^	14.33 ± 0.577 ^bc^	31.00 ± 1.167 ^hi^	12.16 ± 0.142 ^c^
Yellow honeydew melon	5	4.10 ± 0.100 ^de^	0.41 ± 0.030 ^a^	5.03 ± 0.208 ^a^	29.80 ± 0.361 ^ab^	14.67 ± 0.577 ^bc^	27.67 ± 0.643 ^gh^	13.12 ± 0.062 ^d^

Mean values ± standard deviations; n = 10; ^a,b,c (…)^—mean values followed by the different letters in the same line differ significantly at the significance level *p* ≤ 0.05.

## Data Availability

Not applicable.

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
