# Peer review of "The Health-Promoting and Sensory Properties of Tropical Fruit Sorbets with Inulin"

_molecules, 2022, doi:10.3390/molecules27134239_

Round 1

Reviewer 1 Report

This is an interesting work. 

Minor comments are:

Abstract: 

The % information (2% and 5 %) corresponding to wt%? Please check  

Check this line “Both the type of fruit and the addition of inulin”. Which type of fruit is about? Because avocado, kiwi, honey melon, yellow melon and mango are fruits 

Check this line “Three types of sorbets were made, two with (2% and 5%) ” It is quite similar to this previous idea: “the effect of the addition of 2% and 5% inulin” 

Precise this line “Sorbets are a good source of antioxidant compounds.” Are all sorbets of this work good sources of antioxidant compounds? 

Introduction: 

Please include references for these lines: 

Page 1, lines 34-40 “Sorbets are low in calories..... Sorbets are low in calories” 

Page 1, lines 41-43 

Page 2, line 48 “As a fat substitute, inulin lowers the energy value of the product.” 

Page 2, line 50 “Sorbets can be a valuable carrier of prebiotics in the diet” 

Page 2, lines 52-53 “Inulin occur... fermented in the colon” 

Page 2, lines 53-56  “Food..... gel integrity.” 

Page 2, lines 62-66 “The speed.... parameters.” 

Page 2, lines 88-90 “Fruits are..... phenolic compounds” 

Page 3, lines 105-107 “The kiwifruit.... gold varieties” 

Page 3, lines 109-114 “The mango... polyphenols.” 

2.1. Raw Materials for an Non-Dairy Ice Cream Dessert: 

Can be change non-dairy ice cream dessert by sorbet? 

Is it possible move to introduction the information of the lines 126-131 (This text: “The basic ingredients of the sorbets were fresh....... antidiabetic properties”) 

Table 1:  

% information corresponding to wt%? Please check 

2.2. Sensory Evaluation:  

Can include more information about the panel composition (male or female; age)? 

Author Response

Dear Reviewer,

We are honored and grateful for reviewing our manuscript. All your suggestions and corrections were very valuable and helpful for us. Manuscript corrected according to all the suggestions was submitted in track-change mode. All modifications were described in our reply below.

Response to reviewer:

[1] The % information (2% and 5 %) corresponded to wt%, that information was added in abstract in line 16 and in chapter 2 (materials and methods).
[2] In line “Both the type of fruit and the addition of inulin” fruit was changed to fruits.
[3] In line “Three types of sorbets were made, two with (2% and 5%) ” It is quite similar to this previous idea: “the effect of the addition of 2% and 5% inulin” - we removed 2% and 5% line 14.
[4] Precise this line “Sorbets are a good source of antioxidant compounds.” Are all sorbets of this work good sources of antioxidant compounds? We changed to “Sorbets can be a good source of antioxidant compounds” due to it was general statement.
[5] Introduction: Please include references for these lines (…) – all references have been included.
[6] 2.1. Raw Materials for an Non-Dairy Ice Cream Dessert: Can be change non-dairy ice cream dessert by sorbet?  Of course, we changed it.
[7] Is it possible move to introduction the information of the lines 126-131 (This text: “The basic ingredients of the sorbets were fresh....... antidiabetic properties”) – yes, we moved it as suggested.
[8] Table 1:  % information corresponding to wt%? Please check wt% - yes it corresponded to wt%, the information was added in the table title.
[9] Sensory Evaluation:  Can include more information about the panel composition (male or female; age)? – The information was added in the paper (7 female, 3 male, aged 30-49).

Once again, we are very thankful for your attention and contribution to the improvement of the manuscript and hope that we met all your expectations with resubmitted article.

Kind Regards

Agnieszka Palka

Reviewer 2 Report

1. This article is very valuable and informative.

2. If possible, please insert some photo to proof your sensory properties.

Author Response

Dear Reviewer,

We are honored and grateful for reviewing our manuscript. Thank you very much for your kind opinion.

Response to reviewer:

[2] If possible, please insert some photo to proof your sensory properties.
Unfortunately we didn’t take any photos during our investigation, although thank you for this suggestion, we will improve our next publications with photographs.

All Reviewers’ suggestions and corrections were very valuable and helpful for us. Manuscript corrected according to all the suggestions was submitted in track-change mode.

Once again, we are very thankful for your attention and contribution to the improvement of the manuscript and hope that we met all your expectations with resubmitted article.

Kind Regards

Agnieszka Palka

Reviewer 3 Report

Dear authors, the paper is mainly based on theoretical research, experimental part is poor and the manuscript is not suitable for Molecules. I strongly suggest the submission to

Dietetics (ISSN 2674-0311) is an international, peer-reviewed, open access journal of human dietetics. It publishes reviews, regular research papers, and communications. Our aim is to publish timely experimental and theoretical research results in a rapid and readily accessible manner.

or

Nutraceuticals (ISSN 1661-3821) is an international, peer-reviewed, open-access journal on the discovery, development, and production of nutraceuticals. It publishes reviews, original research papers, communications, case reports, and short notes. Our aim is to encourage scientists to publish their theoretical and experimental results in as much detail as possible. There is no restriction on the length of papers. Full experimental and/or methodical details must be provided.

Author Response

Dear Reviewer,

We are honored and grateful for reviewing our manuscript. All your suggestions and corrections were very valuable and helpful for us. Manuscript corrected according to all Reviewers’ suggestions was submitted in track-change mode. We submitted the paper to Molecules, believing that the proposed topics and research methods fit into the nature and specificity of the journal. Thank you very much for your suggestion and in the future we would like to send our next manuscript to Dietetics (ISSN 2674-0311) or Nutraceuticals (ISSN 1661-3821).

Once again, we are very thankful for your attention and contribution to the improvement of the manuscript and hope that we met all your expectations with resubmitted article.

Kind Regards

Agnieszka Palka

Reviewer 4 Report

I have read the manuscript “The health-promoting and sensory properties of tropical fruit sorbets with the addition of inulin” by Agnieszka Palka and Magdalena Skotnicka (MS # molecules-1774182) submitted for the publication in Molecules.

In their manuscript the authors investigated the effect of the substitution of few percents of fruit with inulin on the sensory and chemical physics properties of some tropical fruit sorbets.

The topic is interesting but, it is opinion of the referee, that the manuscript needs major revisions before its publication in Molecules.

In particular:

1.      The authors did not add inulin to sorbets, but substituted fruit with inulin, so the term addition should be changed in the title and all over the text.

2.      As the authors reported, the properties of sorbets are strongly affected by the used cultivars, degree of fruit ripeness, and time elapsed since fruit picking. Therefore, the authors should not stress too much the difference between sorbets made with different fruits. On the contrary, it is more relevant the difference between the sorbets made with the same fruit.

3.      A score rate for sensory evaluation of the sorbets present in Table 2 is mandatory. How was the final total score (line 381) given? Just an arithmetic average?

4.      All the chemical physical properties reported in Table 3 should be renormalized to the real amount of fruit present in the sorbets to correct the difference (that the same authors reported) in acidity, Vitamin C, polyphenols and DPPH. In addition, this renormalization could affect the results of their statistical analysis.

5.      Line 131: the purity degree of used inulin should be reported or estimated.

6.      Being inulin slightly sweet, can the substitution of fruit with inulin affect the taste of sorbets as a consequence of the more or less sweetness of the substituted fruit?

7.      Line 146 and followings: The term melting point is misunderstanding. It could be better to use melting parameter or something similar.

8.      Line 385: the last sentence should be rephrased as it cannot be considered a result of the present research. On the contrary, the sensory evaluation reported for avocado sorbets was the worst as reported in line 382.

9.      There some apexes missing all over the text.

Author Response

Dear Reviewer,

We are honored and grateful for reviewing our manuscript. All your suggestions and corrections were very valuable and helpful for us. Manuscript corrected according to all the suggestions was submitted in track-change mode. All modifications were described in our reply below.

Response to reviewer:

[1] The authors did not add inulin to sorbets, but substituted fruit with inulin, so the term addition should be changed in the title and all over the text.
The title was changed by removing “the addition of” – we used term “addition” instead of “substitution” because the addition would always be a substitution. Sorbets are made of fruits and water as main ingredients, therefore addition of any stabilizer, fibre or sugar would be a substitution, depending on amount and type of replaced base ingredient (in this case – fruit). Nevertheless we explained in the paper (in 2.2. Preparation of the Sorbets), that “addition of inulin” was in fact “replacement of fruits” in the same amount.

[2] As the authors reported, the properties of sorbets are strongly affected by the used cultivars, degree of fruit ripeness, and time elapsed since fruit picking. Therefore, the authors should not stress too much the difference between sorbets made with different fruits. On the contrary, it is more relevant the difference between the sorbets made with the same fruit.
The addition of inulin (both 2 and 5%wt.) did not statistically affect organoleptic features of the sorbets. There were differences in other parameters and it was described in more detail in the paper.

[3] A score rate for sensory evaluation of the sorbets present in Table 2 is mandatory.
As given in 2.2. 5-point hedonic scale was used and score rate was given in Table 2 (added as suggested).

[4] All the chemical physical properties reported in Table 3 should be renormalized to the real amount of fruit present in the sorbets to correct the difference (that the same authors reported) in acidity, Vitamin C, polyphenols and DPPH. In addition, this renormalization could affect the results of their statistical analysis.
Our aim was to evaluate given parameters in the whole product, which were sorbets with 2 different amounts of inulin and without inulin. Therefore renormalization to real amount in the fruit would make the aim of this study impossible to achieve. We agree, that the characterization of fruits properties would be crucial, therefore we want to continue our study and improve it with the determination of the parameters of fresh fruits.

[5] Line 131: the purity degree of used inulin should be reported or estimated.
The inulin was chicory inulin, purchased in local store, as described in the study. We didn’t estimate its purity, because it was not the aim of our study. According to the information given on the package, the product was produced in Belgium and contained 89g of fiber per 100g of product. In our future research we will verify the purity of inulin.

[6] Being inulin slightly sweet, can the substitution of fruit with inulin affect the taste of sorbets as a consequence of the more or less sweetness of the substituted fruit?
The inulin sweetness is about 1/10 of sugar. The addition of 2 and 5 wt.% is probably too small to change the taste of the final product. Nevertheless we would like to estimate this parameter in our future studies.

[7] Line 146 and followings: The term melting point is misunderstanding. It could be better to use melting parameter or something similar.
In the case of all kinds ice cream the term “melting” and “melting point” are commonly used. We changed this in this study into “melting resistance” and explained in chapter 2.1 its meaning.

[8] Line 385: the last sentence should be rephrased as it cannot be considered a result of the present research. On the contrary, the sensory evaluation reported for avocado sorbets was the worst as reported in line 382.
That sentence and conclusions were rephrased.

Once again, we are very thankful for your attention and contribution to the improvement of the manuscript and hope that we met all your expectations with resubmitted article.

Kind Regards

Agnieszka Palka

Round 2

Reviewer 3 Report

I have exactly the same comments of the first revision, ear authors, the paper is mainly based on theoretical research, experimental part is poor and the manuscript is not suitable for Molecules. I strongly suggest the submission to

Dietetics (ISSN 2674-0311) is an international, peer-reviewed, open access journal of human dietetics. It publishes reviews, regular research papers, and communications. Our aim is to publish timely experimental and theoretical research results in a rapid and readily accessible manner.

or

Nutraceuticals (ISSN 1661-3821) is an international, peer-reviewed, open-access journal on the discovery, development, and production of nutraceuticals. It publishes reviews, original research papers, communications, case reports, and short notes. Our aim is to encourage scientists to publish their theoretical and experimental results in as much detail as possible. There is no restriction on the length of papers. Full experimental and/or methodical details must be provided.

Reviewer 4 Report

The authors have made the essential corrections, provided some detailed answers to some of the questions, and postponed some minor comments for future investigations. Overall the manuscript was improved and can be published in Molecules as it is.